# Sex biased effect of acute heat shock on the antioxidant system of non-native round goby *Neogobius melanostomus*

**Dagmara Błońska**[1]*, **Bartosz Janic**[1], **Ali Serhan Tarkan**[1,2], **Bożena Bukowska**[3]

**1** Department of Ecology and Vertebrate Zoology, Faculty of Biology and Environmental Protection, University of Łódź, Łódź, Poland, **2** Department of Basic Sciences, Faculty of Fisheries, Muğla Sıtkı Koçman University, Menteşe, Muğla, Turkey, **3** Department of Biophysics of Environmental Pollution, Faculty of Biology and Environmental Protection, University of Łódź, Łódź, Poland

* dagmara.blonska@biol.uni.lodz.pl

**Data Availability Statement:** The authors have uploaded their underlying data to Dryad: Błońska, Dagmara (2021), Miniatura data, Dryad, Dataset, https://doi.org/10.5061/dryad.m37pvmd3h

## Abstract

Monitoring oxidative stress biomarkers has become a powerful and common tool to estimate organismal condition and response to endogenous and environmental factors. In the present study, we used round goby (*Neogobius melanostomus*) from non-native European populations, as a model species to test sex differences in oxidative stress biomarkers. Considering sex differences in reproductive investment, we hypothesized that males would display lower resistance to abiotic stress. Fish were exposed to a heat shock (temperature elevated by 10˚C) for 1h, 6h, and 12h and catalase activity (CAT), reduced glutathione (GSH), total antioxidant capacity (TAC) and lipid peroxidation (LPO) were measured in liver and muscle tissues. Liver of males was significantly more responsive compared to liver of females in all tested parameters. GSH was found to be the most responsive to heat stress exposure in both sexes. The results supported our hypothesis that male reproductive investment (territoriality, courtship, and brood care) and likelihood of only a single spawning period in their lifetime influenced on higher sensitivity of their antioxidant defence. On the other hand, for females antioxidant defence is considered more important to survive the environmental changes and successfully reproduce in the next season. Our experiments exposed fish to acute thermal stress. Further research should determine the effects of exposure to chronic thermal stress to corroborate our understanding on sex differences in antioxidant defence in the round goby.

## 1. Introduction

Temperature is one of the key environmental factors affecting an organism's physiological processes, especially in ectotherms, which adjust their metabolism to the ambient temperature. Although aquatic organisms are acclimated to daily and seasonal variations in temperature [1], sudden and unexpected increase can substantially influence their overall performance [2, 3]. This includes long-term challenges arising from climate change but also from human-mediated thermal pollution, such as cooling water in thermal plant stations [4]. Forecasted global

(https://datadryad.org/stash/share/
2QpSiudLFJSWThN8j-jFk8BZy_wig5GKTKqy21_
BAY8).

**Funding:** The study was founded by the National
Science Centre (project no. 2018/02/X/NZ8/
00056). The funders had no role in study design,
data collection and analysis, decision to publish, or
preparation of the manuscript.

**Competing interests:** The authors have declared
that no competing interests exist.

warming will result in a temperature increase at 1.5 ˚C between 2030 and 2052, however, not all regions will be faced with the same heat [5]. The temperature increase due to anthropogenic global warming is a constant, progressive process estimated to enhance the air temperature at 0.2 ˚C per decade [5]. In contrast, extreme unpredictable events such as heat waves, can modify the thermal regime significantly at short term and persist for approximately 8–11 days [6]. The intensity, duration, and frequency of such events are expected to increase [6]. Freshwater ecosystems vary in their susceptibility to temperature fluctuations, which is generally related to their size and depth. Alterations of the river's temperature regime due to thermal pollution and global warming impact both native and non-native species, with non-native species having higher potential to benefit from such changes [7]. In this regard, freshwater ecosystems, among all waterbodies [8], are one of the most vulnerable to climate change at all latitudes [7, 9, 10].

Exposure of aerobic organisms to environmental changes elicits responses to maintain homeostasis and biological functions. Adjustments at different stages are involved (e.g., genetic, physiological), however, before they reach an observable level (e.g., behavioural, morphological), they can be sensed within the cell. From the set of various environmental stressors, changes in temperature levels are considered substantial stress factor influencing metabolic rate and leading to oxidative stress [e.g., 11, 12]. The imbalance between antioxidants and oxidants, in favour of the latter (oxidative stress), may influence cellular constituents' modification and disturbance of cellular metabolism [13]. Organisms respond to such a situation and develop efficient defence comprising of low and high molecular mass antioxidants as well as antioxidant enzymes [14]. Although most of them are well known, their level and activity (in the case of enzymes) are usually species-specific, may differ between populations, ontogenetic stages, or gender [14, 15]. In recent years, monitoring of oxidative stress parameters became a powerful tool enabling the evaluation of organism condition, including fish [16]. However, Rudneva & Skuratouskaya [17] suggested that there was little attention paid to the physiological and natural factors in contrast to anthropogenic factors in biomonitoring and environmental studies. While some of them are easily mitigated, e.g., by collecting samples at the same time of the season, others (such as sex and age) require further studies [17].

In accordance with the increase of temperature in Europe in the last decades, the expansion of non-native gobiids was observed (e.g., in the Danube River; [18]). These fish species, originating from the Ponto-Caspian region, evolved in the harsh continental climate, which has most likely shaped their wide tolerance to various environmental factors [7] and contributed to their rapid dispersion through European waters as well as North America [19]. Among them, the round goby *Neogobius melanostomus* was included in the 100 worst invaders in Europe [20] as well as is regarded as one of the most wide-ranging invasive fish on Earth [19]. Its success results from many features as it displays wide tolerance to abiotic factors, opportunistic diet, aggressive behaviour, and effective reproductive strategy [19]. Given many studies conducted on its successful establishment in different waterbodies within Europe and North America makes the species an attractive model to study [19]. Kovyrshina and Rudneva [21–23] previously demonstrated that round goby displays an adaptive response to oxidative stress caused by various factors connected with anthropogenic load and seasonality.

In our study, we investigated the acute thermal shock influence on round goby to test intersexual differences in the effectiveness of antioxidant defence. Many oxidative stress studies conducted on fish have not considered 'sex' factor [24–27]. Some studies have not indicated any differences [e.g., 17], while others have confirmed that it is an important factor affecting oxidative stress parameters [e.g., 28, 29]. The spawning cycle of the round goby is well known and was documented under laboratory conditions [30]. In the pre-spawning period, male searches for a suitable place for reproduction, guards it, and starts building the nest, while

females appear later to lay eggs [31]. Egg inspection and ventilation, as well as aggressive chasing of intruders, are the main activities of a male until the eggs hatch. During the breeding season, males usually do not forage [19, 30] and spawn more frequently compared to females (i.e., more reproductive acts per individual) [32]. Considering the reproductive activities displayed by males (territoriality, courtship, and brood care), the burden affecting males during the long breeding period may be higher than the expenditure of females [32]. It was suggested that males usually die after their first reproductive season [31]. Thus, we hypothesised that high and extended reproductive investment of round goby males should translate into weaker defence mechanisms associated with oxidative stress compared to females. We used elevated temperature as a stressor, because this factor has high probability to affect gobies due to both warming climate [7], especially heat waves, and thermal pollution [4]. The 10 ˚C increase was used to ensure heat shock, which may reflect thermal pollution associated with the release of cooling water from thermal plant stations [4] or extreme heat waves [33] hitting more often due to climate warming.

## 2. Materials and methods

### 2.1. Fish

Round goby specimens were collected using electrofishing (type EFGI 650, BES Bretschneider Spezialelektornik, Germany) in September 2018 in the Radunia River in Pruszcz Gdański (permission obtained from water tenant—Polish Angling Association in Gdańsk L. Dz.611/19), Poland (54˚16'50"N, 18˚38'22"E). Mature individuals, ranged between 89–144 mm in total length and weighed 8.14–33.17 g, of both sexes (gender determination is easy to conduct in the field based on the shape of urogenital papillae, [31]) were collected. Sampled fish were transported in aerated tanks to the laboratory, and after 24 h of acclimation, they were divided by sex, placed in 70 L aquaria (4–5 individuals) equipped with halves of PVC pipes to provide shelter (5 cm long, exceeding the number of fish to avoid competition). To ensure proper living conditions, all aquaria were connected in a flow-through system (the same volume exchanged constantly). The light regime was set 12 d:12 n to reflect natural conditions. The temperature in the laboratory room was maintained at 18–19 ˚C. Fish were fed every second day with frozen chironomid larvae and kept in such conditions for 4 weeks, which enabled fish to acclimate to laboratory conditions and level the condition of both sexes after reproduction. For all procedures, permission from the Local Ethics Committee was obtained (41/LB102/2018).

### 2.2. Experimental setup and protocol

Fish were tested in 45 L non-transparent, mildly aerated tanks equipped with a single shelter (similar to those used in stocking aquaria) and aquarium heater. Before the experimental trials, each individual was kept for 24 h separately in the tank to acclimate to the experimental conditions (water temperature 18–19 ˚C). Such conditions were also used as a separate, control treatment (KC; N = 12 → 6 females and 6 males). After this period, fish were moved to similar tanks with heated water (29–30 ˚C) for 1 h, 6 h, or 12 h (KT1, KT2, KT3, respectively—three separate treatments; in each N = 12 → 6 females and 6 males) to mimic acute heat stress. According to Lee & Johnson [34], the upper thermal limit for the species was established to be around 29 ˚C, with a critical temperature above 33 ˚C (reviewed in Kornis et al. [19]). Then, fish were killed by spinal cord rupture, liver and muscle tissues, which are commonly used and easy to obtain in relatively high amount, were removed and immediately frozen in a temperature below -80 ˚C for further analyses.

## 2.3. Biochemical analysis

Sampled tissues (liver, muscle) were homogenised using a X-120 knife homogeniser (CAT Ingenieurbüro GmbH, Germany) in 100 mM sodium phosphate buffer (pH 7.4, 100 mM KCl, 1 mM $Na_2$-EDTA) with 100 μM PMSF dissolved in ethanol (98%). Homogenization was performed on ice at 3500 rpm for 4 min, and the homogenates were then centrifuged at 4 ˚C for 10 min (15 000 rpm). Then, the supernatants were removed for the following estimations. The total protein determination was based on Lowry et al. [35] method. For each sample, three technical replicate measurements were taken.

To assess the antioxidant defence, we measured the level of reduced glutathione (GSH) and the activity of catalase (CAT). To determine the level of GSH, we performed the modified Ellman's method [36], where we added 20% trichloroacetic acid (TCA) to the homogenates (final concentration 2%) and centrifuged the samples (15 000 rpm, 10 min). To the obtained supernatant, we added 10 mM DTNB ((5,5'-dithiobis (2-nitrobenzoic acid)) and 500 mM sodium phosphate buffer. After 20 min of incubation in darkness, due to the formation of yellow 5-thio-2-nitrobenzoate ion corresponding to the GSH concentration, the absorbance of this ion was measured at 412 nm, calculated and expressed as a μmol GSH/mg protein in homogenate. Molar absorption coefficient (ε) for DTNB is 13.6 x $10^3$ $M^{-1}cm^{-1}$. The CAT activity was determined based on the enzyme ability to decompose $H_2O_2$ [37]. Homogenates were added to the solution consisting of 54 mM $H_2O_2$ in 50 mM potassium phosphate buffer (pH 7.00). The degradation of hydrogen peroxide was measured during 1 min at 240 nm. As one unit of CAT activity, the amount of enzyme is taken, which decomposes 1 μmol of hydrogen peroxide during 1 min. CAT activity in homogenates was calculated in μmol $H_2O_2$/min/mg protein in the homogenate.

The general activity of the antioxidant mechanism was measured using total antioxidant capacity (TAC), a method based on the reduction of 2,2'-azino-bis(3-ethylbenzothiazoline-6-sulphonic acid) (ABTS) [38]. Measurements were performed at 414 nm, and TAC was expressed in one-electron equivalents of Trolox. Considering that one Trolox molecule reacts with two $ABTS^+$ molecules, the calculated values were multiplied by two (units per μmol of Trolox equivalent $L^{-1}$). Molar absorption coefficient (ε) for ABTS is 36 x $10^3$ $M^{-1}$ $cm^{-1}$.

To evaluate the oxidative damage, the concentration of the end product of lipid peroxidation, malondialdehyde (MDA), was measured [39]. Homogenate was added to the solution of 20% TCA, 0,6% thiobarbituric acid (TBA) in HCl (36–38%), shaken, and then centrifuged (3000 rpm, 5 min). The supernatant was heated in 100 ˚C for 15 min, cooled, mixed with *n*-buthanol, and centrifuged (3000 rpm, 5 min). The spectrophotometric measurements of MDA levels were performed in the buthanol phase at 532 nm, expressed as nmol MDA/mg protein in the homogenate. Molar absorption coefficient (ε) for MDA is 1.56 x $10^5$ $M^{-1}$ $cm^{-1}$.

## 2.4. Statistics

The obtained data were calculated per total protein concentration. The outliers were determined using +/- two standard deviations and removed, which in some cases decreased the number of replicates from six to three. Differences in oxidative stress parameters between tissue, treatment and sex were examined by permutational univariate analysis of variance (PERANOVA) using PERMANOVA+ v.1.0.1 for PRIMER version 6.1.11 (PRIMER-E LTD, Plymouth, UK). Data analysis was based on a three-way (fully-crossed) design, which included the fixed factors (Tissue, Treatment, and Sex). Using an Euclidean distance measure following normalisation of the data, these were used to obtain a distance matrix, which was subjected to 9,999 permutations of the raw data and tested for significance, with a posteriori pairwise comparisons evaluated at α = 0.05. Briefly, the advantage of PERANOVA compared with

traditional parametric analysis of variance is that the stringent assumptions of normality and homoscedasticity in the data, which prove very often unrealistic when dealing with ecological datasets, are significantly relaxed [40], and has widely been used in several previous similarly designed studies [41–43].

## 3. Results

### 3.1. GSH

Significant Tissue × Treatment × Sex interaction ($F^{\#} = 4.7906$, $P^{\#} = 0.0052$; $^{\#}$ = permutational value) indicated differences in the level of GSH between sexes depending on tissue and treatment. In liver, along with the time of heat exposure, the GSH level in male tissues increased up to 6h with sharp reduction in the 12h treatment (KT2 > KC; $t^{\#} = 5.9168$, $P^{\#} = 0.0016$ and KT2 < KT3; $t^{\#} = 4.022$, $P^{\#} = 0.0085$; Fig 1). GSH levels in female livers were similar in all heat shock treatments. Comparison of sex showed that after 6 and 12 hours of heat exposure, GSH level in male livers was significantly higher compared to females ($t^{\#} = 10.233$, $P^{\#} = 0.0002$ and $t^{\#} = 3.8682$, $P^{\#} = 0.0084$, respectively). The opposite pattern was observed in muscle where differences between sexes were recorded after 1h and 6h of heat exposure ($t^{\#} = 3.3204$, $P^{\#} = 0.0197$ and $t^{\#} = 6.838$, $P^{\#} = 0.0003$, respectively; Fig 1). On the contrary to the pattern observed in the liver, male muscle tissues showed that it decreased from KC to KT2 ($t^{\#} = 6.7964$, $P = 0.0004$) and from KT1 to KT2 ($t^{\#} = 4.5692$, $P^{\#} = 0.0035$) with a sudden increase in value in the 12h treatment (KT3 > KT2, $t^{\#} = 4.9291$, $P = 0.0017$). Female muscles were more responsive to heat shock than liver. After a sharp decrease of GSH level from KC to KT1 ($t^{\#} = 4.7978$, $P^{\#} = 0.0016$), GSH increased in the subsequent treatments KT2 and KT3 (KT1 < KT2, $t^{\#} = 5.59$, $P^{\#} = 0.001$; KT1 < KT3, $t^{\#} = 6.0002$, $P^{\#} = 0.0005$).

### 3.2. CAT

The level of CAT differed between sexes in Tissue × Sex interaction ($F^{\#} = 35.932$, $P^{\#} = 0.0001$) but only in the liver ($t^{\#} = 5.493$, $P^{\#} = 0.0001$; Fig 2).

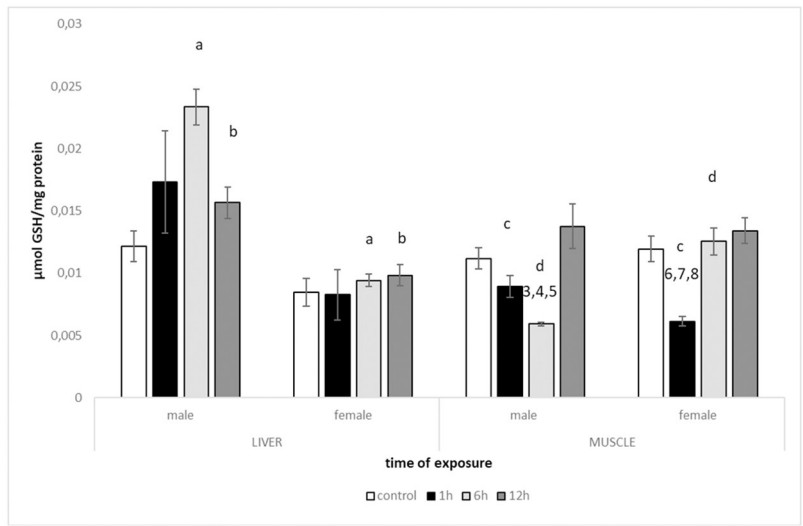

**Fig 1. Levels of reduced glutathione GSH (mean +/- SE) measured in tissues of males and females of round goby exposed to heat shock (+10˚C) for 1h, 6h and 12h (N = 3–6).** Significant differences between males and females within the same tissues were marked with letters (a-d) and significant differences between treatments were marked with numbers (1–8) (p < 0. 005).

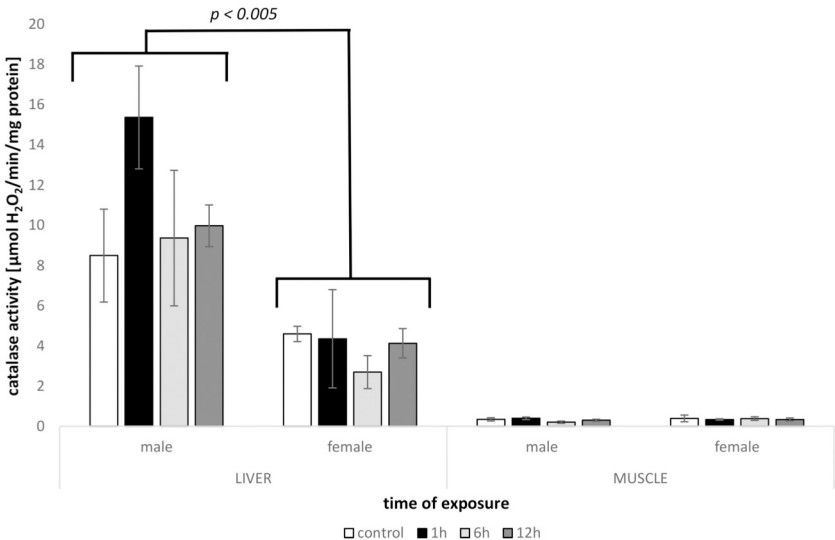

**Fig 2. Levels of catalase activity CAT (mean +/- SE) measured in tissues of males and females of round goby exposed to heat shock (+10˚C) for 1h, 6h and 12h (N = 4–6).** Groups with CAT levels significantly different are marked (p < 0. 005).

## 3.3. TAC

Significant Tissue × Treatment and Tissue × Sex interaction ($F^{\#}$ = 3.0272, $P^{\#}$ = 0.0358; $F^{\#}$ = 34.815, $P^{\#}$ = 0.0001; respectively) indicated differences in total antioxidant capacity between sexes and treatments depending on the tissue. Posteriori pairwise comparisons confirmed differences between sexes in the liver ($t^{\#}$ = 4.9432, $P^{\#}$ = 0.0001), but not in the muscle ($t^{\#}$ = 1.2298, $P^{\#}$ = 0.2209). In the male liver, the level of TAC increased just after 1h, was kept high up to 6h, then sharply decreased after 12h of heat exposure. However, this pattern was not statistically significant (Fig 3). The general differences among treatments in liver tissue were significant in comparison with KC versus KT2 ($t^{\#}$ = 2.4265, $P^{\#}$ = 0.0328) and KT2 versus KT3 ($t^{\#}$ = 2.6342, $P^{\#}$ = 0.0226).

## 3.4. Oxidative damage—Lipid peroxidation

The oxidative damage was also different between sexes, demonstrated by significant Tissue × Sex and Treatment × Sex interaction ($F^{\#}$ = 25.647, $P^{\#}$ = 0.0001, $F^{\#}$ = 3.2654; $P^{\#}$ = 0.0288; respectively). These differences were significant in posteriori pairwise comparisons between males and females in the liver ($t^{\#}$ = 4.5762, $P^{\#}$ = 0.0001), but not in the muscle ($t^{\#}$ = 1.8475, $P^{\#}$ = 0.0767). Lipid peroxidation in the liver was on a constant level in all heat shock treatments, whereas in males, the level of LPO increased up to 6h and returned below the control value after 12h of heat exposure (Fig 4). In the Treatment × Sex interaction posteriori pairwise test, sex comparisons were significant in the treatment KC ($t^{\#}$ = 3.2625, $P^{\#}$ = 0.0049) and KT2 ($t^{\#}$ = 3.0896, $P^{\#}$ = 0.0082) only.

The general pattern indicated that after 12h of heat exposure, measured antioxidant parameters returned to the initial (control) stage.

## 4. Discussion

The present study suggests diverse responses of males and females of round goby to the oxidative insult evoked by a heat shock. In all tested parameters (CAT, LPO, GSH, TAC), liver of

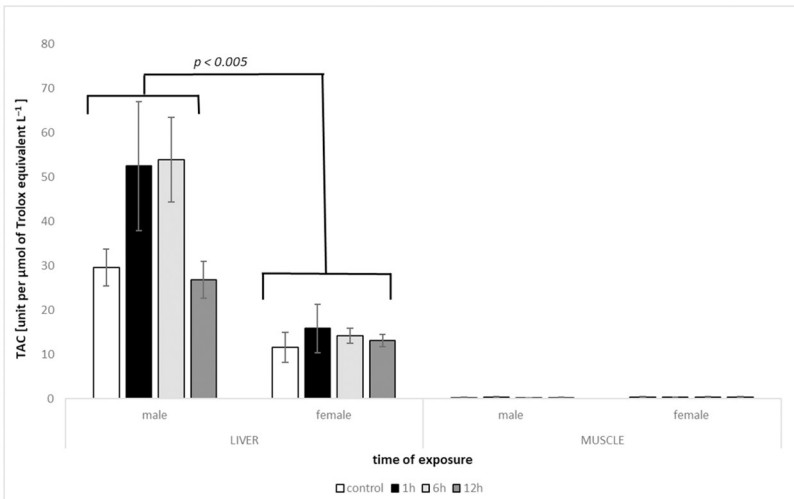

**Fig 3. Levels of total antioxidant capacity TAC (mean +/- SE) measured in tissues of males and females of round goby exposed to heat shock (+10˚C) for 1h, 6h and 12h (N = 4–6).** Groups with TAC levels significantly different are marked (p < 0.005).

males displayed greater responsiveness compared to females, which did not exhibit substantial changes during the whole experimental procedure. The oxidative defence was measured via catalase activity and reduced glutathione levels, commonly used oxidative stress biomarkers (reviewed in Birnie-Gauvin et al. [15]). Although the fluctuation of CAT activity in females in both tested tissues was recorded, the differences among different times of heat exposure were not significant. Similar situation was indicated for males, despite even up to two-fold increase (after 1 h in liver) or decrease (after 6 h in muscle) of catalase activity in heated water.

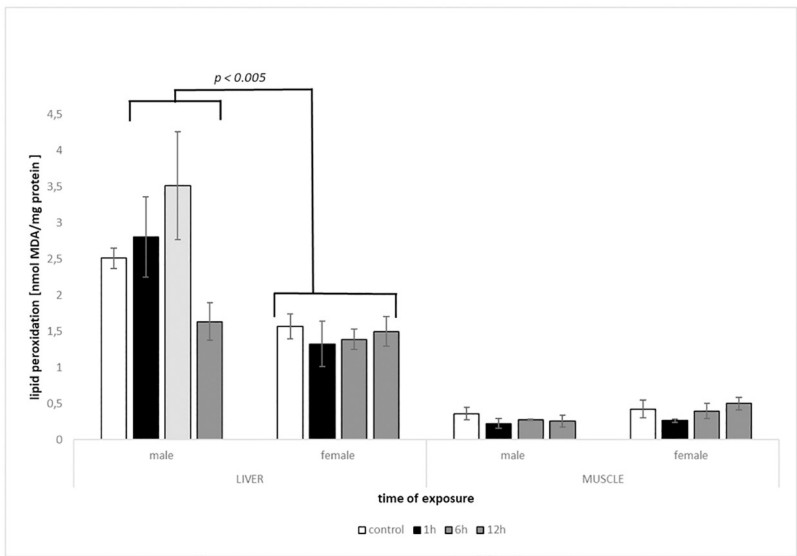

**Fig 4. Levels of lipid peroxidation LPO (mean +/- SE) measured in tissues of males and females of round goby exposed to heat shock (+10˚C) for 1h, 6h and 12h (N = 3–6).** Groups with LPO levels significantly different are marked (p < 0. 005).

Bagnyukova et al. [25] suggested high sensitivity and thermoinactivation of catalase in *Perccottus glenii* Amur sleeper (in liver tissue), but the results of our study showed activity of the enzyme, which did not decrease more than 15% below the control level. Due to the high metabolic rate of the liver, it could be expected that CAT activity would be enhanced, especially considering that the enzyme is the most effective, dealing with oxidative stress, when the level of $H_2O_2$ is highly elevated [44]. Other studies, where fish were exposed to the temperature beyond their thermal optimum evidenced the significant effect on catalase activity [e.g., 45, 46]. However, small concentrations of $H_2O_2$ are supposed to be controlled by glutathione peroxidases [44], a group of enzymes which use reduced glutathione as a substrate to decompose hydrogen peroxide and display a higher affinity for $H_2O_2$ [47]. In our study, GSH level turned out to be the most responsive to heat stress exposure. The main supplier of GSH is liver [44, 47], which in this case showed an enhanced value, however, not significant in females. Increase in GSH in the liver as a response to heat shock corresponded with the decrease of this parameter in muscle tissue (both sexes). It is possible that via the transport network GSH demands in muscle were met by liver supply, which can be evidenced in males by the increased values of GSH in 12 h treatment in muscle with a simultaneous decrease in liver. Female round goby tissues responded in a different pattern with enhanced values observed after 6 h of heat exposure (muscle); however, without substantial changes in GSH level in liver. The reduced glutathione level is also the first line in ROS inactivation, involved in the detoxification of many endogenous compounds and xenobiotics [47]. This feature enables conjugation with GSH, which may be frequent in accelerated metabolism due to elevated temperature. Total antioxidant capacity was used as a general parameter, exhibiting the antioxidant potential of homogenates. It also confirmed the higher sensitivity of male's liver to thermal stress compared to females, while muscles of both sexes did not show any substantial changes. All antioxidant defences measured returned close to the basal level, which suggests enhanced tolerance of the species to thermal stress [11, 26].

To assess the oxidative damage, MDA content was determined, despite many objections, a method commonly used and relevant in lipid peroxidation evaluation (Hermes-Lima 2004). Again, sex differences were observed in the liver with male tissues being more sensitive compared to females. The muscles of both sexes responded in a similar pattern, a decrease in LPO in the first hour of heat exposure and then an increase in the following treatments. The applied experimental procedure was inspired by the work of Lushchak & Bagnyukowa [48] and Bagnyukowa et al. [25], who tested other fish species also displaying wide tolerance to environmental factors: goldfish *Carassius auratus* and invasive Amur sleeper. In both cases, TBARS levels expressing lipid peroxidation, measured in the liver increased. Muscle tissue showed different pattern in goldfish and Amur sleeper, where in the former increase in TBARS was observed and in the latter enhance after the first hour of heat exposure with a decrease below the control level in the following treatments. In mentioned studies, sex was not determined [25, 48]. Liver of round goby males showed a similar pattern to goldfish and Amur sleeper, while muscles of both sexes exhibited the opposite one. Other studies, where fish species were exposed to gradual increasing water temperature indicate general enhance in LPO, when temperature rose by 10°C [27, 49].

The aim of our study was to confirm that the male antioxidant mechanisms are less efficient compared to females in round goby. We conducted the experiments at the end of the reproductive period to avoid the direct impact of spawning on oxidative stress parameters. Additionally, the acclimation time was long enough to compensate the reproductive effort and balance condition of both sexes. Conducted studies confirmed our hypothesis that males of round goby will display higher sensitivity to oxidative stress. It is not obvious which sex carries the heavier burden of reproductive effort. In the pre-spawning period, females most of the

energy invest in ovaries and oocyte development. The energy expenditure for egg production increases with female age from 20% in the first year up to 50% in the third [31]. Although reproductive activity requires high resources allocation, after laying their eggs, females return to deeper waters, e.g., to avoid predation [19]. Testes and all the associated structures represent much smaller proportion of male body weight compared to female ovaries; however, the secondary reproductive effort could be greater than the primary one in gobies [32]. Round goby males precede females in moving into shallow waters to seek for and prepare the nest, where they remain and guard it throughout the reproductive season [19, 31]. They also display a higher growth rate (with even 1.5 times greater increments than females) leading to sexual dimorphism [31, 32]. Males' investment in the secondary characters such as visual (body colouration), auditory, and olfactory cues, is greater than females and is linked to territoriality, courtship, and brood care [32]. The number of spawning during the lifetime may be of key importance. It is suggested that males' high expenditures during reproduction contribute to their mortality after the first reproductive season [31, 32], while females reproduce approximately over three years [32]. This could partially explain the difference in both sex response to environmental stress, which in the case of this study was elevated temperature (heat shock). Long activity in most of the year, starting in early spring when males migrate to nearshore waters and finishing in mid-autumn, as well as food deprivation for most males is fatal. Thus, additional investment in oxidative defence might be of low importance and had limited effects on male reproductive success. The opposite refers to females, which enhance their energy expenditure with age and reproduce more than once [32].

## 5. Conclusions

The efficiency of handling oxidative stress is critical for a range of key life-history traits, including reproduction, because the associated physiological processes lead to reactive oxygen species (ROS) generation [50]. The adequate response to an oxidative insult might turn out to be crucial for survival under environmental changes such as seasonality or food availability. Results of our study have ecological relevance from the viewpoint of stress tolerance, which appears to be greater in females. However, we acknowledge that in our study we measured only a part of commonly used oxidative stress parameters (reviewed in Brine-Gauvin et al. [15]) and exposed fish only to acute but not chronic stress. Superoxide dismutase (SOD), glutathione peroxidase (GPX), and glutathione reductase (GR), DNA damage, or protein carbonyl measurements, along with measurement of responses to chronic stress, could substantially enrich our understanding of the complexity of the response to environmental stresses in round goby.

## Acknowledgments

Authors would like to thank Dr Radke and Dr Bernaś for their assistance in the field. We thank Lorenzo Villizi for his help in statistical analysis. The authors declare no conflicts of interest.

## Author Contributions

**Conceptualization:** Dagmara Błońska, Bartosz Janic.

**Formal analysis:** Ali Serhan Tarkan.

**Funding acquisition:** Dagmara Błońska.

**Investigation:** Dagmara Błońska, Bartosz Janic.

**Methodology:** Bożena Bukowska.

**Project administration:** Dagmara Błońska.

**Resources:** Dagmara Błońska, Bartosz Janic, Bożena Bukowska.

**Supervision:** Dagmara Błońska.

**Visualization:** Dagmara Błońska, Ali Serhan Tarkan.

**Writing – original draft:** Dagmara Błońska, Bożena Bukowska.

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
