## [Decision Letter · Decision Letter 0]

15 Sep 2021

PONE-D-21-22718Sex biased effect of acute heat shock on the antioxidant system of non-native round goby Neogobius melanostomusPLOS ONE

Dear Dr. Blonska,

Thank you for submitting your manuscript to PLOS ONE. After careful consideration, we feel that it has merit but does not fully meet PLOS ONE’s publication criteria as it currently stands. Therefore, we invite you to submit a revised version of the manuscript that addresses the points raised during the review process.It is crucial a full revision of statistics and to follow most recommendations of referees.

Please submit your revised manuscript within 40 days. If you will need more time than this to complete your revisions, please reply to this message or contact the journal office at plosone@plos.org. Please include the following items when submitting your revised manuscript:A rebuttal letter that responds to each point raised by the academic editor and reviewer(s). You should upload this letter as a separate file labeled 'Response to Reviewers'.A marked-up copy of your manuscript that highlights changes made to the original version. You should upload this as a separate file labeled 'Revised Manuscript with Track Changes'.An unmarked version of your revised paper without tracked changes. You should upload this as a separate file labeled 'Manuscript'.

We look forward to receiving your revised manuscript.

Kind regards,

Marcelo Hermes-Lima, PhD

Academic Editor

PLOS ONE

Reviewers' comments:

Reviewer's Responses to Questions

**Comments to the Author**

1. Is the manuscript technically sound, and do the data support the conclusions?

Reviewer #1: No

Reviewer #2: Partly

2. Has the statistical analysis been performed appropriately and rigorously? 

Reviewer #1: No

Reviewer #2: Yes

3. Have the authors made all data underlying the findings in their manuscript fully available?

Reviewer #1: Yes

Reviewer #2: No

4. Is the manuscript presented in an intelligible fashion and written in standard English?

Reviewer #1: Yes

Reviewer #2: Yes

5. Review Comments to the Author

Reviewer #1: 1、The authors hypothesized that extended reproductive investment of males should translate into weaker defense mechanisms associated with oxidative stress compared with females. In this case, a comparation of antioxidant systems between mature and juvenile fish for both genders are particularly necessary in the experiment design.

2、I have noticed that only the tissues of muscle and live were included in the experiment, why not use the whole blood or blood serum in your experiment design?

3、Instead of only including a single group of control samples, the control treatment samples should also be collected and examined at 1h, 6h and 12h respectively.

4、Statistical methods for the analysis and presentation of the results does not seem to be correct.

Reviewer #2: Manuscript number: PONE-D-21-22718

Article type: Research Article

Full title: Sex biased effect of acute heat shock on the antioxidant system of non native Round Goby, Neogobiusmelanostomus

Short title: Sex biased response of Round Goby to acute heat shock

Corresponding author: Dagmara Blonska

University of Lodz, Faculty of biology and environmental protection, Lodz, Poland

Keyword: Oxidative stress, Invasive species, Thermal stress, Reproductive effort, Gender differences.

Reviewer Comments

This paper investigates an interesting aspects of redox physiology regarding sex biased effect of acute heat shock in Round Goby. However its current state leaves much to be desired. Antioxidant status in females has been reported to be significantly lower than the males which are not as per the previous findings in many animals. Reproductive investment by males has been told to be more than the female which needs more evidence in its support. Some graphs need symbols clarifying the difference between different groups with appropriate level of significance. While the paper is well written, it needs more explanations about the findings citing more references.

Major concerns:

1. Authors have mentioned “Male reproductive investment and likelihood of only single spawning period in their lifetime does not enable them to develop efficient antioxidant defence”, in the abstract section. However from the graph it seems males to be with higher level of GSH, Catalase and TAC in control condition than the female counter parts. This contradiction should be explained.

2. It has been reported in many animals regarding higher antioxidant levels in females than males. Even sex hormone estrogen shows its antioxidant activity. Here authors have reported an opposite result which should be explained and should be supported with more evidences considering other antioxidant enzymes and non enzymatic antioxidant.

3. Regarding reproductive investment, it was told that the female Round Goby’s reproductive investment is less than the male. It should also be explained with appropriate reason and citation. Usually females invest a lot than the male for reproduction.

4. Authors have shown in the graph regarding higher responses in changes of antioxidant level by male Goby than the female. However no explanation regarding increase, decrease or coming down to the control level have been given with respect to pro-oxidant (MDA) and antioxidant after heat treatment.

5. In Discussion section (246-248) it was told ..........”It is possible that via the transport network GSH demands in muscle were met by liver supply, which can be evidenced in males by the increased value of GSH in 6h treatment in muscle with a simultaneous decrease in liver”. However graph showing GSH data is different i.e. GSH in 6h treatment in muscle of male is decreasing and GSH in 6h treatment in liver of male is increasing. Likewise in the line 253-256 it was told that increase in GSH level after 12h treatment, Coincides with a decrease in MDA levels (but not significant). However graph shows no decrease in MDA level in muscle tissue after 12h treatment.

6. Results as represented in graphs have not been explained considering variations in data due to heat treatment and sex biasedness.

Minor Concerns

1. Abbreviations used in graphs for Significant differences among different group should be clearly mentioned in the caption.

6. PLOS authors have the option to publish the peer review history of their article (what does this mean?). If published, this will include your full peer review and any attached files.

Reviewer #1: No

Reviewer #2: **Yes: **Dr. Debadas Sahoo

---

## [Author Response · Author response to Decision Letter 0]

7 Oct 2021

Reviewer Comments

Reviewer #1: 

1、The authors hypothesized that extended reproductive investment of males should translate into weaker defense mechanisms associated with oxidative stress compared with females. In this case, a comparation of antioxidant systems between mature and juvenile fish for both genders are particularly necessary in the experiment design.

RESPONSE: Reported maturation of round goby was at 2 yr for females and 3 yr for males in their native range (e.g. Marsden et al. 1996). However, in the non-native area round gobies were observed to spawn earlier, with females with developed gonads already at age of 1 (MacInnis & Corkum 2000, Tomczak & Sapota 2006). In the juvenile specimens it is not possible to distinguish between sexes visually (contrary to mature individuals), so it could be difficult to conduct similar studies using juveniles. Comparing juveniles and mature individuals will also be misleading because of the potential age differences (Birnie-Gauvin et al. 2017).

Marsden, J. E., Charlebois, P., Wolfe, K., Jude, D. J., & Rudnicka, S. (1996). The round goby (Neogobius melanostomus): a review of European and North American literature. INHS Center for Aquatic Ecology 1996 (10).

MacInnis, A. J., & Corkum, L. D. (2000). Fecundity and reproductive season of the round goby Neogobius melanostomus in the upper Detroit River. Transactions of the American Fisheries Society, 129(1), 136-144.

Tomczak, M. T., & Sapota, M. R. (2006). The fecundity and gonad development cycle of the round goby (Neogobius melanostomus Pallas 1811) from the Gulf of Gdańsk. Oceanological and Hydrobiological Studies, 35(4), 353-367.

Birnie‐Gauvin K, Costantini D, Cooke SJ, Willmore WG. A comparative and evolutionary approach to oxidative stress in fish: a review. Fish and Fish. 2017; 18: 928-942. https://doi.org/10.1111/faf.12215

2、I have noticed that only the tissues of muscle and live were included in the experiment, why not use the whole blood or blood serum in your experiment design?

RESPONSE: Among different tissues sampled for assessment of oxidative stress parameters, liver and muscle are most often taken and display response to oxidative insult (e.g. Luschak & Bagnyukowa 2006, Bagnyukowa et al. 2007, Venagre et al. 2012, Madeira et al. 2013). The study was inspired by works of Bagnyukowa and Lushchak (2006), where they used various tissues, including liver and muscle, to analyze oxidative stress parameters. To be able to compare obtained results with other studies (close to ours considering the design), we decided to use similar tissues, as the level of particular antioxidants is tissue specific. Additionally, these tissues are possible to be easily collected in relatively high amount, while blood sample would be very limited (it is possible that the amount would be insufficient for all the analysis) and requires much higher precision (included in the MS; lines 137-138).

3、Instead of only including a single group of control samples, the control treatment samples should also be collected and examined at 1h, 6h and 12h respectively.

RESPONSE: We agree with the reviewer that this would be a nice complement to our work. However, in this case, we used control group as a source of basic information on oxidative stress parameters in round goby, without exposition to additional factors. Both sexes were exposed to the same treatment, which enabled us to perform comparison between them.

4、Statistical methods for the analysis and presentation of the results does not seem to be correct.

RESPONSE: We are not sure why the reviewer thought this way (mainly because the reviewer did not raise any particular flaws of the statistical approach used and direct us what should be the right way to do it), as the statistical methods and presentation are all widely used and accepted, evidenced in many previous papers. Nevertheless, we have provided further explanations why this methodology is appropriate and how commonly used before with proper examples (lines 186-189).

Reviewer #2

This paper investigates an interesting aspect of redox physiology regarding sex biased effect of acute heat shock in Round Goby. However its current state leaves much to be desired. Antioxidant status in females has been reported to be significantly lower than the males which are not as per the previous findings in many animals. Reproductive investment by males has been told to be more than the female which needs more evidence in its support. Some graphs need symbols clarifying the difference between different groups with appropriate level of significance. While the paper is well written, it needs more explanations about the findings citing more references.

Major concerns:

1. Authors have mentioned “Male reproductive investment and likelihood of only single spawning period in their lifetime does not enable them to develop efficient antioxidant defence”, in the abstract section. However from the graph it seems males to be with higher level of GSH, Catalase and TAC in control condition than the female counter parts. This contradiction should be explained.

RESPONSE: We have reconsidered this statement and changed it (see lines 27-28). We have tried to highlight higher sensitivity of male’s antioxidant response than females, which we believe is connected with their reproductive strategy. Differences in the basic level of tested parameters are not significant. Although we see that the level of particular parameters is higher in males, at this stage we cannot explain them. It could be possible that males display higher steady-state of ROS concentration than females.

2. It has been reported in many animals regarding higher antioxidant levels in females than males. Even sex hormone estrogen shows its antioxidant activity. Here authors have reported an opposite result which should be explained and should be supported with more evidences considering other antioxidant enzymes and non enzymatic antioxidant.

RESPONSE: We agree with the reviewer. In many animals, females are also more burden considering reproductive investment than males and display higher antioxidant level. However, in case of gobies, their specific reproductive strategy is recognized with its own name “tokology”, which include both the mechanics and the profitability of reproduction (Miller 1984), i.e. due to high costs for both genders. Although females transfer energy into egg development, which occur 2-3 times during the spawning (batch spawners), in the meantime they retreat into deeper waters to avoid predation and develop next batch of oocytes. They invest less into growth, as their body increments are 1.5 less than males (Charlebois et al. 1997). As we mentioned in the manuscript, testes of males involve less energy investment, however all secondary activities are highly energy consuming (lines 316-318). They also constantly protect the nest and eggs for a couple of months, deprived of food, which most probably cause their mortality after first spawning (lines 318-326). A negative correlation of oxidative stress resistance and parental care was shown e.g., in smallmouth bass Micropterus dolomieu (Wilson et al. 2012).

In our study we indicated higher sensitivity of goby male antioxidant system in response to acute stress. Although basic (control) levels of tested parameters were higher in males (especially in liver), these particular differences were not significant. Explaining this discrepancy is beyond us without additional research. We agree that more evidence would complete our study, however, there are several approaches of analysing of oxidative stress as well as various enzymatic and non-enzymatic antioxidants to test. We decided for one of the most commonly used.

Wilson, S. M., Gravel, M., Mackie, T. A., Willmore, W. G., & Cooke, S. J. (2012). Oxidative stress associated with paternal care in smallmouth bass (Micropterus dolomieu). Comparative Biochemistry and Physiology Part A, 162, 212–218. 

3. Regarding reproductive investment, it was told that the female Round Goby’s reproductive investment is less than the male. It should also be explained with appropriate reason and citation. Usually females invest a lot than the male for reproduction.

RESPONSE: We tried to explain comprehensively the differences in reproduction of males and females in round goby, both in the introduction (93-102) and discussion (lines 311-327). In case of gobies, Miller (1984) distinguished two kinds of reproductive effort. The primary one is connected with biosynthesis and storage within the gametes, which is most noticeable in oocyte size and number. The secondary is divided into “(a) the anabolism of secondary sexual characters, including dimorphism in growth, body proportions, finnage, colouration, and perhaps in pheromone output, and (b) the catabolism of reproductive behaviour, involving the work done in territoriality, nidification, courtship, spawning and brood care, as well as any preliminary migration to breeding areas”. Considering the great involvement of males into the second sector of the reproductive effort (e.g., greater body increments, dark colouration, courtship, active and aggressive defence of the nest, constant egg ventilation) and the duration of spawning (pre-spawning migration, seeking and holding the nest, eggs protection, several spawning events with different females), it is not obvious that female’s investment is higher than males, which we mentioned in the discussion. We believe also that possibility for multiple spawning in females contrary to single spawning in males is crucial for the antioxidant mechanisms both sexes display.

4. Authors have shown in the graph regarding higher responses in changes of antioxidant level by male Goby than the female. However no explanation regarding increase, decrease or coming down to the control level have been given with respect to pro-oxidant (MDA) and antioxidant after heat treatment.

RESPONSE: We indeed highlighted at the end of the result that the general pattern was returning to the initial (control) level all tested parameters (lines 233-234) as well as in the discussion (283-285).

5. In Discussion section (246-248) it was told ..........”It is possible that via the transport network GSH demands in muscle were met by liver supply, which can be evidenced in males by the increased value of GSH in 6h treatment in muscle with a simultaneous decrease in liver”. However graph showing GSH data is different i.e. GSH in 6h treatment in muscle of male is decreasing and GSH in 6h treatment in liver of male is increasing. Likewise in the line 253-256 it was told that increase in GSH level after 12h treatment, Coincides with a decrease in MDA levels (but not significant). However graph shows no decrease in MDA level in muscle tissue after 12h treatment.

RESPONSE: We agree with the reviewer and thank for these remarks. In the cited fragment of the text should be 12 h treatment, not 6 h, and we have changed that in the MS (lines 260-261). According to the second comment, we have removed this sentence.

6. Results as represented in graphs have not been explained considering variations in data due to heat treatment and sex biasedness.

RESPONSE: Apologies if we have misunderstood this comment but results represented in the graphs are already explained in the text (in the Results section) – they are all connected. 

Minor Concerns

1. Abbreviations used in graphs for Significant differences among different group should be clearly mentioned in the caption.

RESPONSE: OK, we have made sure that ‘significant’ are mentioned in the figure captions.

---

## [Decision Letter · Decision Letter 1]

15 Nov 2021

Sex biased effect of acute heat shock on the antioxidant system of non-native round goby Neogobius melanostomus

PONE-D-21-22718R1

Dear Dr. Blonska,

We’re pleased to inform you that your manuscript has been judged scientifically suitable for publication and will be formally accepted for publication once it meets all outstanding technical requirements.

Kind regards,

Marcelo Hermes-Lima, PhD

Academic Editor

PLOS ONE

Additional Editor Comments (optional):

Reviewers' comments:

Reviewer's Responses to Questions

**Comments to the Author**

1. If the authors have adequately addressed your comments raised in a previous round of review and you feel that this manuscript is now acceptable for publication, you may indicate that here to bypass the “Comments to the Author” section, enter your conflict of interest statement in the “Confidential to Editor” section, and submit your "Accept" recommendation.

Reviewer #1: All comments have been addressed

Reviewer #2: All comments have been addressed

2. Is the manuscript technically sound, and do the data support the conclusions?

Reviewer #1: Yes

Reviewer #2: Yes

3. Has the statistical analysis been performed appropriately and rigorously? 

Reviewer #1: Yes

Reviewer #2: Yes

4. Have the authors made all data underlying the findings in their manuscript fully available?

Reviewer #1: Yes

Reviewer #2: No

5. Is the manuscript presented in an intelligible fashion and written in standard English?

Reviewer #1: Yes

Reviewer #2: Yes

6. Review Comments to the Author

Reviewer #1: I have no further comments to the authors. The authors seem to have addressed the comments, so I recommend the manuscript to be accepted for publication.

Reviewer #2: Authors have modified the manuscript at desired key points as instructed. However responses from male and female towards heat treatment with respect to oxidative stress should summarised from different angles prior to writing breif conclusion.

7. PLOS authors have the option to publish the peer review history of their article (what does this mean?). If published, this will include your full peer review and any attached files.

Reviewer #1: No

Reviewer #2: No

---

## [Editor Report · Acceptance letter]

7 Dec 2021

PONE-D-21-22718R1 

Sex biased effect of acute heat shock on the antioxidant system of non-native round goby *Neogobius melanostomus*

Dear Dr. Błońska:

I'm pleased to inform you that your manuscript has been deemed suitable for publication in PLOS ONE. Congratulations! Your manuscript is now with our production department. 

Kind regards, 

on behalf of

Dr. Marcelo Hermes-Lima 

Academic Editor

PLOS ONE